# D4AM: A General Denoising Framework for Downstream Acoustic Models

**Chi-Chang Lee**[1,2]**, Yu Tsao**[2]**, Hsin-Min Wang**[2]**, Chu-Song Chen**[1,2]
[1]National Taiwan University, Taipei, Taiwan
[2]Academia Sinica, Taipei, Taiwan

## ABSTRACT

The performance of acoustic models degrades notably in noisy environments. Speech enhancement (SE) can be used as a front-end strategy to aid automatic speech recognition (ASR) systems. However, existing training objectives of SE methods are not fully effective at integrating speech-text and noisy-clean paired data for training toward unseen ASR systems. In this study, we propose a general denoising framework, D4AM, for various downstream acoustic models. Our framework fine-tunes the SE model with the backward gradient according to a specific acoustic model and the corresponding classification objective. In addition, our method aims to consider the regression objective as an auxiliary loss to make the SE model generalize to other unseen acoustic models. To jointly train an SE unit with regression and classification objectives, D4AM uses an adjustment scheme to directly estimate suitable weighting coefficients rather than undergoing a grid search process with additional training costs. The adjustment scheme consists of two parts: gradient calibration and regression objective weighting. The experimental results show that D4AM can consistently and effectively provide improvements to various unseen acoustic models and outperforms other combination setups. Specifically, when evaluated on the Google ASR API with real noisy data completely unseen during SE training, D4AM achieves a relative WER reduction of 24.65% compared with the direct feeding of noisy input. To our knowledge, this is the first work that deploys an effective combination scheme of regression (denoising) and classification (ASR) objectives to derive a general pre-processor applicable to various unseen ASR systems. Our code is available at https://github.com/ChangLee0903/D4AM.

## 1 INTRODUCTION

Speech enhancement (SE) aims to extract speech components from distorted speech signals to obtain enhanced signals with better properties (Loizou, 2013). Recently, various deep learning models (Wang et al., 2020; Lu et al., 2013; Xu et al., 2015; Zheng et al., 2021; Nikzad et al., 2020) have been used to formulate mapping functions for SE, which treat SE as a regression task trained with noisy-clean paired speech data. Typically, the objective function is formulated using a signal-level distance measure (e.g., L1 norm (Pandey & Wang, 2018; Yue et al., 2022), L2 norm (Ephraim & Malah, 1984; Yin et al., 2020; Xu et al., 2020), SI-SDR (Le Roux et al., 2019; Wisdom et al., 2020; Lee et al., 2020), or multiple-resolution loss (Défossez et al., 2020)). In speech-related applications, SE units are generally used as key pre-processors to improve the performance of the main task in noisy environments. To facilitate better performance on the main task, certain studies focus on deriving suitable objective functions for SE training.

For human-human oral communication tasks, SE aims to improve speech quality and intelligibility, and enhancement performance is usually assessed by subjective listening tests. Because large-scale listening tests are generally prohibitive, objective evaluation metrics have been developed to objectively assess human perception of a given speech signal (Rix et al., 2001; Taal et al., 2010; Jensen & Taal, 2016; Reddy et al., 2021). Perceptual evaluation of speech quality (PESQ) (Rix et al., 2001) and short-time objective intelligibility (STOI) (Taal et al., 2010; Jensen & Taal, 2016) are popular objective metrics designed to measure speech quality and intelligibility, respectively. Recently, DNSMOS (Reddy et al., 2021) has been developed as a non-instructive assessment tool that predicts human ratings (MOS scores) of speech signals. In order to obtain speech signals with

improved speech quality and intelligibility, many SE approaches attempt to formulate objective functions for SE training directly according to speech assessment metrics (Fu et al., 2018; Fu et al., 2019; 2021). Another group of approaches, such as deep feature loss (Germain et al., 2019) and HiFi-GAN (Su et al., 2020), propose to perform SE by mapping learned noisy latent features to clean ones. The experimental results show that the deep feature loss can enable the enhanced speech signal to attain higher human perception scores compared with the conventional L1 and L2 distances.

Another prominent application of SE is to improve automatic speech recognition (ASR) in noise (Seltzer et al., 2013; Weninger et al., 2015b; Li et al., 2014; Cui et al., 2021). ASR systems perform sequential classification, mapping speech utterances to sequences of tokens. Therefore, the predictions of ASR systems highly depend on the overall structure of the input utterance. When regard to noisy signals, ASR performance will degrade significantly because noise interference corrupts the content information of the structure. Without modifying the ASR model, SE models can be trained separately and "universally" used as a pre-processor for ASR to improve recognition accuracy. Several studies have investigated the effectiveness of SE's model architecture and objective function in improving the performance of ASR in noise (Geiger et al., 2014a; Wang et al., 2020; Zhang et al., 2020; Chao et al., 2021; Meng et al., 2017; Weninger et al., 2015a; Du et al., 2019; Kinoshita et al., 2020; Meng et al., 2018). The results show that certain specific designs, including model architecture and input format, are favorable for improving ASR performance. However, it has also been reported that improved recognition accuracy in noise is not always guaranteed when the ASR objective is not considered in SE training (Geiger et al., 2014b). A feasible approach to tune the SE model parameters toward the main ASR task is to prepare the data of (noisy) speech-text pairs and backpropagate gradients on the SE model according to the classification objective provided by the ASR model. That is, SE models can be trained on a regression objective (using noisy-clean paired speech data) or/and a classification objective (using speech-text paired data). Ochiai et al. (2017a;b) proposed a multichannel end-to-end (E2E) ASR framework, where a mask-based MVDR (minimum variance distortionless response) neural beamformer is estimated based on the classification objective. Experimental results on CHiME-4 (Jon et al., 2017) confirm that the estimated neural beamformer can achieve significant ASR improvements under noisy conditions. Meanwhile, Chen et al. (2015) and Ma et al. (2021) proposed to train SE units by considering both regression and classification objectives, and certain works (Chen et al., 2015; Ochiai et al., 2017a;b) proposed to train SE models with E2E-ASR classification objectives. A common way to combine regression and classification objectives is to use weighting coefficients to combine them into a joint objective for SE model training.

Notwithstanding promising results, the use of combined objectives in SE training has two limitations. First, how to effectively combine regression and classification objectives remains an issue. A large-scale grid search is often employed to determine optimal weights for regression and classification objectives, which requires exhaustive computational costs. Second, ASR models are often provided by third parties and may not be accessible when training SE models. Moreover, due to various training settings in the acoustic model, such as label encoding schemes (e.g., word-piece (Schuster & Nakajima, 2012), byte-pair-encoding (BPE) (Gage, 1994; Sennrich et al., 2016), and character), model architectures (e.g., RNN (Chiu et al., 2018; Rao et al., 2017; He et al., 2019; Sainath et al., 2020), transformer (Vaswani et al., 2017; Zhang et al., 2020), and conformer (Gulati et al., 2020)), and objectives (e.g., Connectionist Temporal Classification (CTC) (Graves et al., 2006), Attention (NLL) (Chan et al., 2016), and their hybrid version (Watanabe et al., 2017)), SE units trained according to a specific acoustic model may not generalize well to other ASR systems. Based on the above limitations, we raise the question: *Can we effectively integrate speech-text and noisy-clean paired data to develop a denoising pre-processor that generalizes well to unseen ASR systems?*

In this work, we derive a novel denoising framework, called D4AM, to be used as a "universal" pre-processor to improve the performance of various downstream acoustic models in noise. To achieve this goal, the proposed framework focuses on preserving the integrity of clean speech signals and trains SE models with regression and classification objectives jointly. By using the regression objective as an auxiliary loss, we circumvent the need to require additional training costs to grid search the appropriate weighting coefficients for the regression and classification objectives. Instead, D4AM applies an adjustment scheme to determine the appropriate weighting coefficients automatically and efficiently. The adjustment scheme is inspired by the following concepts: (1) we attempt to adjust the gradient yielded by a proxy ASR model so that the SE unit can be trained to improve the general recognition capability; (2) we consider the weighted regression objective as a regularizer and, thereby,

prevent over-fitting while training the SE unit. For (1), we derive a coefficient $\alpha_{gclb}$ (abbreviation for gradient calibration) according to whether the classification gradient conflicts with the regression gradient. When the inner product of the two gradient sets is negative, $\alpha_{gclb}$ is the projection of the classification gradient on the regression gradient; otherwise, it is set to 0. For (2), we derive a coefficient $\alpha_{srpr}$ (abbreviation for surrogate prior) based on an auxiliary learning method called ARML (auxiliary task reweighting for minimum-data learning) (Shi et al., 2020) and formulate the parameter distribution induced by the weighted regression objective as the surrogate prior when training the SE model. From the experimental results on two standard speech datasets, we first notice that by properly combining regression and classification objectives, D4AM can effectively improve the recognition accuracy of various unseen ASR systems and outperform the SE models trained only with the classification objective. Next, by considering the regression objective as an auxiliary loss, D4AM can be trained efficiently to prevent over-fitting even with limited speech–text paired data. Finally, D4AM mostly outperforms grid search. The main contribution of this study is two-fold: (1) to the best of our knowledge, this is the first work that derives a general denoising pre-processor applicable to various unseen ASR systems; (2) we deploy a rational coefficient adjustment scheme for the combination strategy and link it to the motivation for better generalization ability.

## 2 MOTIVATION AND MAIN IDEA

**Noise Robustness Strategies for ASR.** Acoustic mismatch, often caused by noise interference, is a long-standing problem that limits the applicability of ASR systems. To improve recognition performance, multi-condition training (MCT) (Parihar et al., 2004; Rajnoha, 2009; Zhang et al., 2020) is a popular approach when building practical ASR systems. MCT takes noise-corrupted utterances as augmented data and jointly trains an acoustic model with clean and noisy utterances. In addition to noise-corrupted utterances, enhanced utterances from SE models can also be included in the training data for MCT training. Recently, Prasad et al. (2021) studied advanced MCT training strategies and conducted a detailed investigation of different ASR fine-tuning techniques, including freezing partial layers (Zhang et al., 2020), multi-task learning (Saon et al., 2017; Prasad et al., 2021), adversarial learning (Serdyuk et al., 2016; Denisov et al., 2018; Zhang et al., 2020), and training involving enhanced utterances from various SE models (Braithwaite & Kleijn, 2019; Zhang et al., 2020; Défossez et al., 2020). The results reveal that recognition performance in noise can be remarkably improved by tuning the parameters in the ASR system. However, an ASR system is generally formed by a complex structure and estimated with a large amount of training data. Also, for most users, ASR systems provided by third parties are black-box models whose acoustic model parameters cannot be fine-tuned to improve performance in specific noisy environments. Therefore, tuning the parameters in an ASR system for each specific test condition is usually not a feasible solution for real-world scenarios.

In contrast, employing an SE unit to process noisy speech signals to generate enhanced signals, which are then sent to the ASR system, is a viable alternative for externally improving the recognition accuracy in noise as a two-stage strategy of robustness. The main advantage of this alternative is that SE units can be pre-prepared for direct application to ASR systems without the need to modify these ASR systems or to know the system setups. Furthermore, ASR is usually trained with a large amount of speech-text pairs, requiring expensive annotation costs; on the contrary, for SE, the required large number of noisy-clean training pairs can be easily synthesized. In this study, we propose to utilize the regression objective as a regularization term, combined with the classification objective, for SE unit training to reduce the need for speech-text data and improve the generalization ability.

**Regression Objective as an Auxiliary Loss.** Since our proposed framework focuses on improving ASR performance in noise, the classification objective is regarded as the main task, and the regression objective is used as an auxiliary loss. Generally, an auxiliary loss aims to improve the performance of the main task without considering the performance of the auxiliary task itself. The auxiliary task learning framework searches for adequate coefficients to incorporate auxiliary information and thus improve the performance of the main task. Several studies (Hu et al., 2019; Chen et al., 2018; Du et al., 2018; Lin et al., 2019; Dery et al., 2021; Shi et al., 2020) have investigated and confirmed the benefits of auxiliary task learning. In D4AM, given the SE model parameter $\theta$, we consider the classification objective $\mathcal{L}_{cls}(\theta)$ as the main task and the regression objective $\mathcal{L}_{reg}(\theta)$ as the auxiliary task. Then, we aim to jointly fine-tune the SE model with $\mathcal{L}_{cls}(\theta)$ and $\mathcal{L}_{reg}(\theta)$, where

their corresponding training data are $\mathcal{T}_{cls}$ (speech-text paired data) and $\mathcal{T}_{reg}$ (noisy-clean paired speech data), respectively. Here, D4AM uses $-\log p(\mathcal{T}_{cls}|\theta)$ and $-\log p(\mathcal{T}_{reg}|\theta)$, respectively, as the classification objective $\mathcal{L}_{cls}(\theta)$ and the regression objective $\mathcal{L}_{reg}(\theta)$. Fig. 1a shows the overall training flow of D4AM. Notably, D4AM partially differs from conventional auxiliary task learning algorithms, where the main and auxiliary tasks commonly share the same model parameters. In D4AM, the main task objective $\mathcal{L}_{cls}(\theta)$ cannot be obtained with only the SE model. Instead, a proxy acoustic model $\phi$ is employed. That is, the true classification objective $\mathcal{L}_{cls}(\theta)$ is not attainable, and alternatively, with the proxy acoustic model $\phi$, a surrogate term $-\log p(\mathcal{T}_{cls}|\theta, \phi)$, termed $\mathcal{L}_{cls}^{\phi}(\theta)$, is used to update the SE parameters. Since there is no guarantee that reducing $\mathcal{L}_{cls}^{\phi}(\theta)$ will reduce $\mathcal{L}_{cls}(\theta)$, a constraint is needed to guide the gradient of $\mathcal{L}_{cls}^{\phi}(\theta)$ toward improving general recognition performance.

Meanwhile, we believe that, for any particular downstream acoustic recognizer, the best enhanced output produced by the SE unit from any noisy speech input is the corresponding clean speech. Therefore, any critical point of $\mathcal{L}_{cls}(\theta)$ should be "covered" by the critical point space of $\mathcal{L}_{reg}(\theta)$, and we just need guidance from speech-text paired data to determine the critical point in the $\mathcal{L}_{reg}(\theta)$ space. That is, when an update of SE model parameters $\theta$ decreases both $\mathcal{L}_{cls}^{\phi}(\theta)$ and $\mathcal{L}_{reg}(\theta)$, that update should be beneficial to general ASR systems. On the other hand, compared to $\mathcal{L}_{cls}(\theta)$, the calculation of $\mathcal{L}_{reg}(\theta)$ does not require the proxy model when we use it to update SE model parameters. Accordingly, we derive the update of $\theta$ at the $t$-th step as:

$$\theta^{t+1} = \arg\min_{\theta} \mathcal{L}_{cls}^{\phi}(\theta) \quad \text{subject to} \quad \mathcal{L}_{reg}(\theta) \leq \mathcal{L}_{reg}(\theta^t). \tag{1}$$

As indicated in Lopez-Paz & Ranzato (2017) and Chaudhry et al. (2019), the model function is supposed to be locally linear around small optimization steps. For ease of computation, we replace $\mathcal{L}_{reg}(\theta) \leq \mathcal{L}_{reg}(\theta^t)$ with $\langle \nabla \mathcal{L}_{cls}^{\phi}(\theta^t), \nabla \mathcal{L}_{reg}(\theta^t) \rangle \geq 0$. That is, we take the alignment between $\nabla \mathcal{L}_{cls}^{\phi}(\theta^t)$ and $\nabla \mathcal{L}_{reg}(\theta^t)$ as the constraint to guarantee the generalization ability to unseen ASR systems at each training step. When the constraint is violated, $\nabla \mathcal{L}_{cls}^{\phi}(\theta^t)$ needs to be calibrated. Following the calibration method proposed in Chaudhry et al. (2019) to search for the non-violating candidate closest to the original $\nabla \mathcal{L}_{cls}^{\phi}(\theta^t)$, we formulate the calibrated gradient $g^*$ as:

$$g^* = \arg\min_{g} \frac{1}{2}\|g - \nabla \mathcal{L}_{cls}^{\phi}(\theta^t)\|^2 \quad \text{subject to} \quad \langle g, \nabla \mathcal{L}_{reg}(\theta^t) \rangle \geq 0. \tag{2}$$

In addition to guiding SE unit training toward improving ASR performance, D4AM also aims to alleviate over-fitting in the classification task. In ARML (Shi et al., 2020), auxiliary task learning was formulated as a problem of minimizing the divergence between the surrogate prior and true prior. It is shown that when a suitable surrogate prior is identified, the over-fitting issue caused by the data limitation of the main task can be effectively alleviated. Following the concept of ARML, D4AM aims to estimate an appropriate coefficient of $\mathcal{L}_{reg}$ to approximate the log prior of the main task objective $\mathcal{L}_{cls}$ to alleviate over-fitting. Finally, D4AM combines the calibrated gradient $g^*$ and auxiliary task learning (ARML) to identify a critical point that ensures better generalization to unseen ASR systems and alleviates over-fitting in the classification task. In Sec. 3, we illustrate the details of our proposed framework.

## 3 METHODOLOGY

D4AM aims to estimate an SE unit that can effectively improve the recognition performance of general downstream ASR systems. As shown in Fig. 1a, the overall objective consists of two parts, the classification objective $\mathcal{L}_{cls}^{\phi}(\theta)$ given the proxy model $\phi$ and the regression objective $\mathcal{L}_{reg}(\theta)$. As shown in Fig. 1b, we intend to guide the update of the model from $\theta_i$ to $\theta_o$ with the regression objective $\mathcal{L}_{reg}(\theta)$ to improve the performance of ASR in noise. In this section, we present an adjustment scheme to determine a suitable weighting coefficient to combine $\mathcal{L}_{cls}^{\phi}(\theta)$ and $\mathcal{L}_{reg}(\theta)$ instead of a grid search process. The adjustment scheme consists of two parts. First, we calibrate the direction of $\nabla \mathcal{L}_{cls}^{\phi}(\theta)$ to promote the recognition accuracy, as described in Eq. 2. Second, we follow the ARML algorithm to derive a suitable prior by weighting the regression objective. Finally, the gradient descent of D4AM at the $t$-th step for SE training is defined as:

$$\theta^{t+1} \leftarrow \theta^t - \epsilon_t(\nabla \mathcal{L}_{cls}^{\phi}(\theta^t) + (\alpha_{gclb} + \alpha_{srpr}) \cdot \nabla \mathcal{L}_{reg}(\theta^t)) + \eta_t, \tag{3}$$

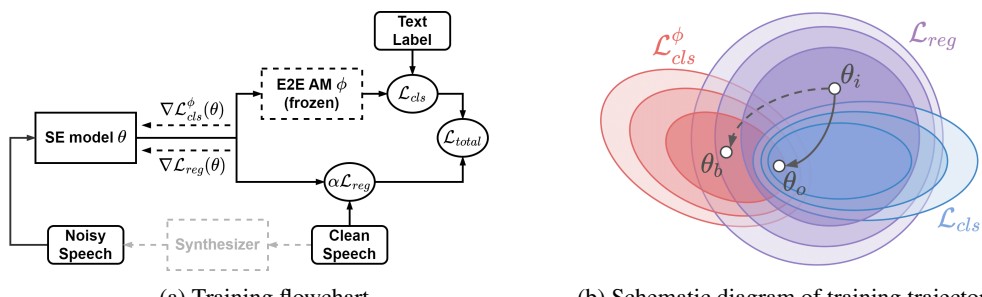

(a) Training flowchart  (b) Schematic diagram of training trajectory

Figure 1: (a) Training flow of D4AM. Given a proxy acoustic model (AM) parameterized by $\phi$, $\theta$ is updated by training with both $\mathcal{L}_{cls}$ and $\mathcal{L}_{reg}$ weighted by $\alpha$. (b) Illustration of the training trajectories. The dashed line (from $\theta_i$ to $\theta_b$) represents the training process without considering $\mathcal{L}_{reg}$. The solid line (from $\theta_i$ to $\theta_o$) represents the training process considering $\mathcal{L}_{reg}$.

where $\epsilon_t$ is the learning rate, and $\eta_t \sim \mathcal{N}(0, 2\epsilon_t)$ is a Gaussian noise used to formulate the sampling of $\theta$. The coefficient $\alpha_{gclb}$ is responsible for calibrating the gradient direction when $\mathcal{L}_{cls}^{\phi}(\theta^t)$ does not satisfy $\langle \nabla \mathcal{L}_{cls}^{\phi}(\theta^t), \nabla \mathcal{L}_{reg}(\theta^t) \rangle \geq 0$. Mathematically, $\alpha_{gclb}$ is the result of projecting $\nabla \mathcal{L}_{cls}^{\phi}(\theta^t)$ into the space formed by using $\nabla \mathcal{L}_{reg}(\theta^t)$ as a basis. The connection between the projection process and the constraint $\langle \nabla \mathcal{L}_{cls}^{\phi}(\theta^t), \nabla \mathcal{L}_{reg}(\theta^t) \rangle \geq 0$ is introduced in Section 3.1. The coefficient $\alpha_{srpr}$ is responsible for weighting $\mathcal{L}_{reg}(\theta^t)$ to approximate the log prior of parameters and is estimated to minimize the divergence between the parameter distributions of the surrogate and true priors. The determination of $\alpha_{srpr}$, the sampling mechanism for $\theta$, and the overall framework proposed in ARML are described in Section 3.2. Finally, we introduce the implementation of the D4AM framework in Section 3.3.

## 3.1 GRADIENT CALIBRATION FOR IMPROVING GENERAL RECOGNITION ABILITY

This section details the gradient calibration mechanism. First, we assume that any critical point of $\mathcal{L}_{cls}(\theta)$ is located in the critical point space of $\mathcal{L}_{reg}(\theta)$. When the derived $g^*$ satisfies $\langle g^*, \nabla \mathcal{L}_{reg}(\theta^t) \rangle \geq 0$, the direction of $g^*$ can potentially yield recognition improvements to general acoustic models. The constrained optimization problem described in Eq. 2 can be solved in a closed form while considering only a mini-batch of samples. We then design a criterion $\mathcal{C} = \langle \nabla \mathcal{L}_{cls}^{\phi}(\theta^t), \nabla \mathcal{L}_{reg}(\theta^t) \rangle$ to examine the gradient direction and formulate the solution as:

$$g^* = \nabla \mathcal{L}_{cls}^{\phi}(\theta^t) + \alpha_{gclb} \cdot \nabla \mathcal{L}_{reg}(\theta^t) \text{ , where } \alpha_{gclb} = -[\![\mathcal{C} < 0]\!] \cdot \frac{\mathcal{C}}{\|\nabla \mathcal{L}_{reg}(\theta^t)\|_2^2}, \quad (4)$$

where $[\![.]\!]$ is an indicator function whose output is 1 if the argument condition is satisfied; otherwise, the output is 0. $\alpha_{gclb}$ is derived based on the Lagrangian of the constrained optimization problem. The details of the derivation are presented in Appendix A.7.

## 3.2 REGRESSION OBJECTIVE WEIGHTING AS THE SURROGATE PRIOR

In this section, we discuss the relationship between the classification and regression objectives with the main ASR task. A weighted regression objective is derived to approximate the log prior for SE training by modifying the ARML (Shi et al., 2020) algorithm, which considers the weighted sum of auxiliary losses as a surrogate prior and proposes a framework for reweighting auxiliary losses based on minimizing the divergence between the parameter distributions of the surrogate and true priors. We aim to identify a coefficient $\alpha_{srpr}$ such that $\alpha_{srpr} = \arg\min_{\alpha} \mathbb{E}_{\theta \sim p^*}[\log p^*(\theta) - \alpha \cdot \log(p(\mathcal{T}_{reg}|\theta))]$, where $p^*$ is the true prior distribution of $\theta$, and $p^{\alpha_{srpr}}(\mathcal{T}_{reg}|\theta)$ is the surrogate prior corresponding to the weighted regression objective $-\alpha_{srpr} \cdot \log(p(\mathcal{T}_{reg}|\theta))$. However, $p^*(\theta)$ is not practically available. To overcome the unavailability of $p^*(\theta)$, ARML uses an alternative solution for $p^*(\theta)$ and a sampling scheme of $\theta$ in $p^*(\theta)$. For the alternative solution of $p^*(\theta)$, ARML assumes that the main task distribution $p(\mathcal{T}_{cls}|\theta)$ will cover $p^*(\theta)$ since the samples of $\theta$ from $p^*(\theta)$ are likely to induce high values in $p(\mathcal{T}_{cls}|\theta)$. Thus, ARML takes $\log p(\mathcal{T}_{cls}|\theta)$ as the alternative of $\log p^*(\theta)$. For the sampling scheme of $\theta$ in $p^*(\theta)$, for convenience, ARML acquires the derived parameters from the joint loss at

each training step and combines them with Langevin dynamics (Neal et al., 2011; Welling & Teh, 2011) to approximate the sampling of $\theta$ with an injected noise $\eta_t \sim \mathcal{N}(0, 2\epsilon_t)$. Thus, the optimization problem of $\alpha_{srpr}$ becomes $\min_\alpha \mathbb{E}_{\theta^t \sim p^J}[\log(p(\mathcal{T}_{cls}|\theta^t)) - \alpha \cdot \log(p(\mathcal{T}_{reg}|\theta^t))]$, where $p^J$ represents the distribution for sampling $\theta$ from the above described joint training process with respect to the original sampling. More details about the ARML algorithm are available in Shi et al. (2020). Finally, at the $t$-th step, the overall parameter change $\Delta\theta(g)$ of the auxiliary learning framework given an arbitrary gradient $g$ of the main task (ASR) can be written as:

$$\Delta\theta(g) = \epsilon_t(g + \alpha_{srpr} \cdot \nabla\mathcal{L}_{reg}(\theta^t)) + \eta_t. \tag{5}$$

Nonetheless, the aforementioned problem remains: we can obtain only $\nabla\mathcal{L}_{cls}^\phi(\theta^t)$ instead of the ideal ASR task gradient $\nabla\mathcal{L}_{cls}(\theta^t)$. Therefore, the gradient should be calibrated, as described in Section 3.1. To this end, we update $\theta$ by accordingly taking $\Delta\theta(g^*)$, as described in Eq. 3.

---

**Algorithm 1** D4AM (A General Denoising Framework for Downstream Acoustic Models)

---

**Learnable-Parameter:** model parameter $\theta$ (initialized by $\theta^0$); task weight $\alpha_{srpr}$ (initialized by 1)
**Hyper-Parameter:** learning rate of the $t$-th iteration $\epsilon_t$, learning rate of task weight $\beta$
 1: **for** iteration $t = 0$ to $T - 1$ **do**
 2: $\quad \alpha_{gclb} \leftarrow \text{project}(\nabla\mathcal{L}_{cls}^\phi(\theta^t), \nabla\mathcal{L}_{reg}(\theta^t))$
 3: $\quad \theta^{t+1} \leftarrow \theta^t - \epsilon_t(\nabla\mathcal{L}_{cls}^\phi(\theta^t) + (\alpha_{gclb} + \alpha_{srpr}) \cdot \nabla\mathcal{L}_{reg}(\theta^t)) + \eta_t$
 4: $\quad g_\alpha \leftarrow \frac{\partial}{\partial\alpha_{srpr}}\|\nabla\mathcal{L}_{cls}^\phi(\theta^t) + (\alpha_{gclb} - \alpha_{srpr}) \cdot \nabla\mathcal{L}_{reg}(\theta^t)\|_2^2$
 5: $\quad \alpha_{srpr} \leftarrow \alpha_{srpr} - \beta \cdot \text{clamp}(g_\alpha, \max = +1, \min = -1)$
 6: **end for**

---

### 3.3 ALGORITHM

In this section, we detail the implementation of the proposed D4AM framework. The complete algorithm is shown in Algorithm 1. The initial value of $\alpha_{srpr}$ is set to 1, and the learning rate $\beta$ is set to 0.05. We first update $\theta^t$ by the mechanism presented in Eq. 3 and collect $\theta$ samples at each iteration. As described in Section 3.2, $\alpha_{srpr}$ is optimized to minimize the divergence between $p^*(\theta)$ and $p^{\alpha_{srpr}}(\mathcal{T}_{reg}|\theta)$. Referring to ARML (Shi et al., 2020), we use the Fisher divergence to measure the difference between the surrogate and true priors. That is, we optimize $\alpha$ by $\min_\alpha \mathbb{E}_{\theta^t \sim p^J}[\|\nabla\mathcal{L}_{cls}(\theta^t) - \alpha \cdot \nabla\mathcal{L}_{reg}(\theta^t))\|_2^2]$. Because we can obtain only $\mathcal{L}_{cls}^\phi(\theta^t)$, we use the calibrated version $g^*$ as the alternative of $\nabla\mathcal{L}_{cls}(\theta^t)$. Then, the optimization of $\alpha$ becomes $\min_\alpha \mathbb{E}_{\theta^t \sim p^J}[\|\nabla\mathcal{L}_{cls}^\phi(\theta^t) + (\alpha_{gclb} - \alpha) \cdot \nabla\mathcal{L}_{reg}(\theta^t))\|_2^2]$. In addition, we use gradient descent to approximate the above minimization problem, since it is not feasible to collect all $\theta$ samples. Practically, we only have access to one sample of $\theta$ per step, which may cause instability while updating $\alpha_{srpr}$. Therefore, to increase the batch size of $\theta$, we modify the update scheme by accumulating the gradients of $\alpha_{srpr}$. The update period of $\alpha_{srpr}$ is set to 16 steps. Unlike ARML (Shi et al., 2020), we do not restrict the value of $\alpha_{srpr}$ to the range 0 to 1, because $\mathcal{L}_{cls}^\phi$ and $\mathcal{L}_{reg}$ have different scales. In addition, we clamp the gradient of $\alpha_{srpr}$ with the interval $[-1, 1]$ while updating $\alpha_{srpr}$ to increase training stability. Further implementation details are provided in Section 4 and Supplementary Material.

## 4 EXPERIMENTS

### 4.1 EXPERIMENTAL SETUP

The training datasets used in this study include: noise signals from DNS-Challenge (Reddy et al., 2020) and speech utterances from LibriSpeech (Panayotov et al., 2015). DNS-Challenge, hereinafter DNS, is a public dataset that provides 65,303 background and foreground noise signal samples. LibriSpeech is a well-known speech corpus that includes three training subsets: Libri-100 (28,539 utterances), Libri-360 (104,014 utterances), and Libri-500 (148,688 utterances). The mixing process (of noise signals and clean utterances) to generate noisy-clean paired speech data was conducted during training, and the signal-to-noise ratio (SNR) level was uniformly sampled from -4dB to 6dB.

The SE unit was built on DEMUCS (Défossez et al., 2019; 2020), which employs a skip-connected encoder-decoder architecture with a multi-resolution STFT objective function. DEMUCS has been

shown to yield state-of-the-art results on music separation and SE tasks, We divided the whole training process into two stages: pre-training and fine-tuning. For pre-training, we only used $\mathcal{L}_{reg}$ to train the SE unit. At this stage, we selected Libri-360 and Libri-500 as the clean speech corpus. The SE unit was trained for 500,000 steps using the Adam optimizer with $\beta_1 = 0.9$ and $\beta_2 = 0.999$, learning rate 0.0002, gradient clipping value 1, and batch size 8. For fine-tuning, we used both $\mathcal{L}_{reg}$ and $\mathcal{L}_{cls}^{\phi}$ to re-train the SE model initialized by the checkpoint selected from the pre-training stage. We used Libri-100 to prepare the speech-text paired data for $\mathcal{L}_{cls}^{\phi}$ and all the training subsets (including Libri-100, Libri-360, and Libri-500) to prepare the noisy-clean paired speech data for $\mathcal{L}_{reg}$. The SE unit was trained for 100,000 steps with the Adam optimizer with $\beta_1 = 0.9$ and $\beta_2 = 0.999$, learning rate 0.0001, gradient clipping value 1, and batch size 16. For the proxy acoustic model $\phi$, we adopted the conformer architecture and used the training recipe provided in the SpeechBrain toolkit (Ravanelli et al., 2021). The model was trained with all training utterances in LibriSpeech, and the training objective was a hybrid version (Watanabe et al., 2017) of CTC (Graves et al., 2006) and NLL (Chan et al., 2016).

Since the main motivation of our study was to appropriately adjust the weights of the training objectives to better collaborate with the noisy-clean/speech-text paired data, we focused on comparing different objective combination schemes on the same SE unit. To investigate the effectiveness of each component in D4AM, our comparison included: (1) without any SE processing (termed **NOIS**), (2) solely trained with $\mathcal{L}_{reg}$ in the pre-training stage (termed **INIT**), (3) solely trained with $\mathcal{L}_{cls}^{\phi}$ in the fine-tuning stage (termed **CLSO**), (4) jointly trained with $\mathcal{L}_{cls}^{\phi}$ and $\mathcal{L}_{reg}$ by solely taking $\alpha_{gclb}$ in the fine-tuning stage (termed **GCLB**), and (5) jointly trained with $\mathcal{L}_{cls}^{\phi}$ and $\mathcal{L}_{reg}$ by solely taking $\alpha_{srpr}$ in the fine-tuning stage (termed **SRPR**). Note that some of the above schemes can be viewed as partial comparisons with previous attempts. For example, **CLSO** is trained only with the classification objective, the same as (Ochiai et al., 2017a;b); **SRPR**, **GCLB**, and **D4AM** use joint training, the same as (Chen et al., 2015; Ma et al., 2021), but (Chen et al., 2015; Ma et al., 2021) used predefined weighting coefficients. In our ablation study, we demonstrated that our adjustment achieves optimal results without additional training costs compared with grid search. Since our main goal was to prepare an SE unit as a pre-processor to improve the performance of ASR in noise, the word error rate (WER) was used as the main evaluation metric. Lower WER values indicate better performance.

To fairly evaluate the proposed D4AM framework, we used CHiME-4 (Jon et al., 2017) and Aurora-4 (Parihar et al., 2004) as the test datasets, which have been widely used for robust ASR evaluation. For CHiME-4, we evaluated the performance on the development and test sets of the 1-st track. For Aurora-4, we evaluated the performance on the noisy parts of the development and test sets. There are two types of recording process, namely wv1 and wv2. The results of these two categories are reported separately for comparison. For the downstream ASR systems, we normalized all predicted tokens by removing all punctuation, mapping numbers to words, and converting all characters to their upper versions. To further analyze our results, we categorized the ASR evaluation into out-of-domain and in-domain scenarios. In the out-of-domain scenario, we aim to verify the generalization ability of ASR when the training and testing data originate from different corpora. That is, in this scenario, the testing data is unseen to all downstream ASR systems. In the in-domain scenario, we aim to investigate the improvements achievable in standard benchmark settings. Therefore, we set up training and testing data for ASR from the same corpus. Note that ASR systems typically achieve better performance in the in-domain scenarios than in the out-domain scenario (Likhomanenko et al., 2021).

## 4.2 EXPERIMENTAL RESULTS

**D4AM Evaluated with Various Unseen Recognizers in the Out-of-Domain Scenario.** We first investigated the D4AM performance with various unseen recognizers. We prepared these recognizers by the SpeechBrain toolkit with all the training data in LibriSpeech. The recognizers include: (1) a conformer architecture trained with the hybrid loss and encoded by BPE-5000 (termed **CONF**), (2) a transformer architecture trained with the hybrid loss and encoded by BPE-5000 (termed **TRAN**), (3) an RNN architecture trained with the hybrid loss and encoded by BPE-5000 (termed **RNN**), and (4) a self-supervised wav2vec2 (Baevski et al., 2020) framework trained with the CTC loss and encoded by characters (termed **W2V2**). To prevent potential biases caused by language models, **CONF**, **TRAN**, and **RNN** decoded results only from their attention decoders, whereas **W2V2** greedily

Table 1: WER (%) results on CHiME-4 and Aurora-4.

| | CHiME-4 | | | | | | | | | | | | | | | |
| | dt05-real | | | | dt05-simu | | | | et05-real | | | | et05-simu | | | |
| | CONF | TRAN | RNN | W2V2 | CONF | TRAN | RNN | W2V2 | CONF | TRAN | RNN | W2V2 | CONF | TRAN | RNN | W2V2 |
|---|---|---|---|---|---|---|---|---|---|---|---|---|---|---|---|---|
| NOIS | 40.53 | 34.56 | 56.76 | 29.84 | 45.40 | 38.86 | 58.75 | 35.37 | 57.23 | 46.10 | 72.16 | 45.61 | 52.06 | 44.96 | 64.93 | 44.10 |
| INIT | 30.98 | 29.61 | 38.61 | 26.93 | 35.02 | 32.49 | 41.76 | 31.84 | 40.32 | 38.77 | 49.67 | 39.37 | 42.78 | 40.51 | 49.83 | 41.04 |
| CLSO | **26.66** | 27.83 | 35.25 | 24.63 | **29.09** | 29.89 | 38.00 | 29.16 | 34.00 | 35.19 | 46.02 | 36.08 | **35.02** | 36.62 | 45.66 | 37.23 |
| SRPR | 26.88 | 26.73 | 34.83 | 24.51 | 30.12 | 29.22 | 38.73 | 29.86 | 34.53 | 33.93 | 46.08 | 35.21 | 36.68 | 36.55 | 46.27 | 38.80 |
| GCLB | 26.76 | 26.51 | 34.62 | 23.55 | 29.28 | 28.73 | 37.55 | 28.54 | 34.18 | 33.76 | 45.75 | 35.15 | 35.69 | 36.09 | 45.37 | **37.18** |
| D4AM | 26.70 | **26.07** | **34.43** | **23.30** | 29.45 | **28.61** | **37.49** | **28.40** | **33.71** | **33.20** | **45.07** | **34.43** | 35.97 | **35.75** | **45.36** | 37.30 |
| | Aurora-4 | | | | | | | | | | | | | | | |
| | dev-wv1 | | | | dev-wv2 | | | | test-wv1 | | | | test-wv2 | | | |
| | CONF | TRAN | RNN | W2V2 | CONF | TRAN | RNN | W2V2 | CONF | TRAN | RNN | W2V2 | CONF | TRAN | RNN | W2V2 |
| NOIS | 21.69 | 19.66 | 27.36 | 14.44 | 31.56 | 27.96 | 42.12 | 25.47 | 19.27 | 17.64 | 23.98 | 13.60 | 29.77 | 27.54 | 38.91 | 26.37 |
| INIT | 18.97 | 18.24 | 21.71 | 13.59 | 27.85 | 26.16 | 34.58 | 24.38 | 16.76 | 16.25 | 18.45 | 12.26 | 26.55 | 25.23 | 31.98 | 24.65 |
| CLSO | 17.75 | 18.23 | 20.98 | 13.31 | 24.76 | 25.05 | 33.08 | 23.48 | **15.43** | 17.03 | 17.53 | 12.16 | 24.24 | 25.85 | 32.33 | 25.79 |
| SRPR | 17.77 | 17.20 | 20.40 | 12.97 | 24.67 | 23.20 | 31.66 | 21.57 | 15.52 | 15.50 | 16.82 | 11.89 | 23.83 | 23.00 | 29.74 | 22.95 |
| GCLB | **17.70** | 17.16 | 20.35 | 12.91 | 24.65 | 22.99 | 31.73 | 21.92 | 15.51 | 15.66 | **16.82** | 12.11 | 24.51 | 23.19 | 30.39 | 23.36 |
| D4AM | 17.73 | **17.08** | **20.30** | **12.84** | **24.31** | **22.78** | **31.06** | **20.90** | 15.55 | **15.44** | 17.18 | **11.66** | **23.35** | **22.68** | **29.65** | **22.02** |

Table 2: WER (%) results of CHiME-4's baseline ASR system.

| method | NOIS | INIT | CLSO | SRPR | GCLB | D4AM |
|---|---|---|---|---|---|---|
| dt05-real | 11.57 | 10.28 | 7.95 | 8.28 | 8.15 | **7.94** |
| dt05-simu | 12.98 | 11.80 | 10.07 | 10.28 | 9.56 | **9.48** |
| et05-real | 23.71 | 19.91 | 16.28 | 16.33 | 16.32 | **15.66** |
| et05-simu | 20.85 | 20.39 | 18.37 | 18.18 | 16.86 | **16.31** |

decoded results following the CTC rule. Table 1 shows the WER results under different settings. Note that we cascaded **CONF** to train our SE model, so **CONF** is not a completely unseen recognizer in our evaluation. Therefore, we marked the results of **CONF** in gray and paid more attention to the results of other acoustic models. Obviously, **NOIS** yielded the worst results under all recognition and test conditions. **INIT** performed worse than others because it did not consider the ASR objective in the pre-training stage. **CLSO** was worse than **GCLB**, **SRPR**, and **D4AM**, verifying that training with $\mathcal{L}_{cls}^{\phi}$ alone does not generalize well to other unseen recognizers. Finally, **D4AM** achieved the best performance on average, confirming the effectiveness of gradient calibration and using regression objective weighting as a surrogate prior in improving general recognition ability.

**D4AM Evaluated with a Standard ASR System in the In-Domain Scenario.** In this experiment, we selected the official baseline ASR system from CHiME-4, a DNN-HMM ASR system trained in an MCT manner using the CHiME-4 task-specific training data, as the evaluation recognizer. This baseline ASR system is the most widely used to evaluate the performance of SE units. As observed in Table 2, the result of **INIT** is comparable to the state-of-the-art results of CHiME-4. Meanwhile, **D4AM** achieved notable reductions in WER compared with **NOIS**. This trend is similar to that observed in Table 2 and again validates the effectiveness of D4AM.

**D4AM Evaluated with a Practical ASR API.** We further evaluated D4AM using the Google ASR API, a well-known black-box system for users. To prepare a "real" noisy testing dataset, we merged the real parts of the CHiME-4 development and test sets. Table 3 shows the average WER values obtained by the Google ASR API. The table reveals trends similar to those observed in Table 1, but two aspects are particularly noteworthy. First, **NOIS** still yielded the worst result, confirming that all SE units contribute to the recognition accuracy of the Google ASR API on our testing data. Second, **CLSO** was worse than **INIT**, confirming that an SE unit trained only by the classification

Table 3: WER (%) results of the Google ASR API on the CHiME-4 real sets.

| method | NOIS | INIT | CLSO | SRPR | GCLB | D4AM |
|---|---|---|---|---|---|---|
| WER (%) | 35.09 | 28.02 | 29.26 | 26.79 | 26.54 | **26.44** |

objective according to one ASR model is not guaranteed to contribute to other unseen recognizers. In contrast, improved recognition results (confirmed by the improvements produced by **SRPR**, **GCLB**, and **D4AM**) can be obtained by well considering the combination of objectives.

## 4.3 ABLATION STUDIES

**Regression Objective Weighting for Handling Over-fitting.** As mentioned earlier, ARML addresses the over-fitting issue caused by the limited labeled data of the main task by introducing an appropriate auxiliary loss. For D4AM, we considered the regression objective as an auxiliary loss to alleviate the over-fitting issue. To verify this feature, we reduced the speech-text paired data to only 10% (2853 utterances) of the Libri-100 dataset. The WER learning curves of **CLSO** and **D4AM** on a privately mixed validation set within 100,000 training steps are shown in Fig. 2a. The figure illustrates that the learning curve of **CLSO** ascends after 50,000 training steps, which is a clear indication of over-fitting. In contrast, **D4AM** displays a different trend. Fig. 2a confirms the effectiveness of using the regression objective to improve generalization and mitigate over-fitting.

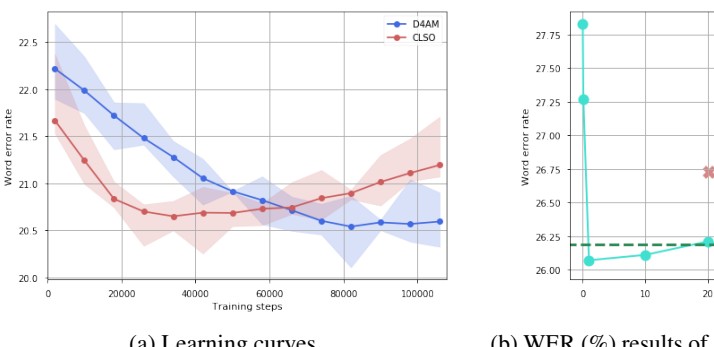

(a) Learning curves.       (b) WER (%) results of grid search, SRPR, and D4AM.

Figure 2: (a) Learning curves of CLSO and D4AM trained with 10% transcribed data of Libri-100. (b) Comparison of WER (%) results of grid search, SRPR, and D4AM. The dashed line represents the average performance of the best seven results of grid search.

**D4AM versus Grid Search.** One important feature of D4AM is that it automatically and effectively determines the weighting coefficients to combine regression and classification objectives. In this section, we intend to compare this feature with the common grid search. Fig. 2b compares the performance of **D4AM**, **SRPR**, and grid search with different weights on the CHiME-4 dt05-real set. For the grid search method, the figure shows the results for different weights, including 0, 0.1, 1.0, 10, 20, 30, 40, 50, and 60. The average performance of the best seven results is represented by the dashed green line. From Fig. 2b, we first note that **D4AM** evidently outperformed **SRPR**. Furthermore, the result of **D4AM** is better than most average results of grid search. The results confirm that D4AM can efficiently and effectively estimate coefficients without an expensive grid search process.

To further investigate its effectiveness, we tested D4AM on high SNR testing conditions, MCT-trained ASR models, and another acoustic model setup. Additionally, we tested human perception assessments using the D4AM-enhanced speech signals. All the experimental results confirm the effectiveness of D4AM. Owing to space limitations, these results are reported in APPENDIX.

## 5 CONCLUSION

In this paper, we proposed a denoising framework called D4AM, which aims to function as a pre-processor for various unseen ASR systems to boost their performance in noisy environments. D4AM employs gradient calibration and regression objective weighting to guide SE training to improve ASR performance while preventing over-fitting. The experimental results first verify that D4AM can effectively reduce WERs for various unseen recognizers, even a practical ASR application (Google API). Meanwhile, D4AM can alleviate the over-fitting issue by effectively and efficiently combining classification and regression objectives without the exhaustive coefficient search. The promising results confirm that separately training an SE unit to serve existing ASR applications may be a feasible solution to improve the performance robustness of ASR in noise.

REPRODUCIBILITY STATEMENT

The source code for additional details of the model architecture, training process, and pre-processing steps, as well as a demo page, are provided in the supplemental material. Additional experiments are provided in Appendix.

ACKNOWLEDGEMENT

This work was partly supported by the National Science and Technology Council, Taiwan (NSTC 111-2634-F-002-023).

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

## A  APPENDIX

### A.1  D4AM EVALUATED ON HIGH SNR DATA

Most SE research focuses on "average" denoising on the target ASR task, which implies that we consider the distortion caused by the SE model as tolerable if it is sufficiently low. Accordingly, we conducted additional experiments using cleaner speech utterances as testing data. The WERs of the Google ASR API are shown in Table 4, where test-clean represents the clean LibriSpeech test set, and the others are test-clean mixed with noisex92 (Varga & Steeneken, 1993) and Nonspeech (Hu & Wang, 2010) noise sources, uniformly sampled at high SNR levels between 10dB to 20dB (average 15 dB). From Table 4 illustrates that the WER of the clean speech processed by D4AM is marginally

higher than that of the original clean speech (ORIG). The result is 16.61% versus 16.27%. However, for both types of high SNR noisy speech, using D4AM reduced the WER (19.24% versus 19.67% for noisex92 (Varga & Steeneken, 1993) and 20.19% versus 20.55% for Nonspeech (Hu & Wang, 2010)). The results show that D4AM can yield performance comparable (better) to that achieved using the original clean (high SNR noisy) speech.

Table 4: WER (%) results on high SNR test sets.

| method | test-clean | test-noisex | test-nonspeech |
|--------|-----------|-------------|----------------|
| ORIG   | 16.27     | 19.67       | 20.55          |
| D4AM   | 16.61     | 19.24       | 20.19          |

## A.2 D4AM EVALUATED WITH MCT-TRAINED ASR MODELS

The results of the Google ASR API in Table 3 are supposed to be representative results of the MCT-trained ASR systems. We conducted additional ASR experiments on CHiME-4 with **MCT-CONF**, **MCT-W2V2**, and **MCT-TRAN** corresponding to the ASR model architectures in Table 1 and trained on the LibriSpeech and DNS noise datasets with the SNR levels -4 to 6. Note that *W2V2* consists of a wav2vec2 module and a linear layer. During training, the wav2vec2 module was fixed and used as a feature extractor, and only the linear layer was updated. It is evident from Table 5 that *D4AM* consistently outperformed *NOIS* for all MCT-trained ASR models.

Table 5: WER (%) results of MCT-trained ASR models on the CHiME-4 dt-05 real set.

| method | MCT-CONF | MCT-W2V2 | MCT-TRAN |
|--------|----------|----------|----------|
| NOIS   | 25.69    | 28.86    | 22.82    |
| D4AM   | 24.22    | 22.89    | 22.59    |

## A.3 D4AM TRAINED WITH ANOTHER ACOUSTIC MODEL

We selected the conformer-based ASR model as the proxy acoustic model because it has a lower memory requirement than other model architectures in SpeechBrain. In terms of implementation, model complexity is a key concern, particularly for researchers similar to us with limited computing resources. We believe our claim that any critical point of $\mathcal{L}_{cls}(\theta)$ should be "covered" by the critical point space of $\mathcal{L}_{reg}(\theta)$ applies to most cases. By considering this appropriately, we can reduce the bias of different settings of the proxy acoustic model used for training. In addition, previous work (Ochiai et al., 2017a;b) has shown empirically that training an SE unit with one acoustic model can improve another ASR system. To show that the proposed D4AM is insensitive to the selection of proxy acoustic model, we conducted additional experiments using the W2V2-CTC model as the proxy acoustic model. The results evaluated with a LibriSpeech-trained ASR system on the CHiME-4 dt-05 real set are shown in Table 6, where Proxy-CONF and Proxy-W2V2-CTC denote conformer- and W2V2-CTC-based proxy acoustic models, respectively.

As shown in Table 6, D4AM can effectively reduce the bias between proxy acoustic models and notably improve the performance. Since Conformer and W2V2-CTC have different model architectures and use different label types, the results confirm that D4AM can consistently produce promising results using different types of proxy acoustic models.

## A.4 D4AM EVALUATED WITH THE L3DAS22 MICRO-A SETS

The input of L3DAS22 Guizzo et al. (2022) is a multi-channel signal. In our experiments, we focused on the single-channel SE task and considered different channels as separate signals for training and testing. Also, please note that the WER definition of L3DAS22 differs from the regular version. In L3DAS22, the reference target for computing WER is **NOT** a true transcription. The WER is the word-level distance between the ASR model's predicted transcripts for the clean and enhanced

Table 6: WER (%) results of different setups on the CHiME-4 dt-05 real set.

| method | CONF | TRAN | RNN |
|---|---|---|---|
| Proxy-CONF-CLSO | 26.66 | 27.83 | 35.25 |
| Proxy-CONF-SRPR | 26.88 | 26.73 | 34.83 |
| Proxy-CONF-GCLB | 26.76 | 27.51 | 34.62 |
| Proxy-CONF-D4AM | 26.7 | 26.07 | 34.43 |
| Proxy-W2V2-CTC-CLSO | 30.61 | 28.98 | 40.30 |
| Proxy-W2V2-CTC-SRPR | 29.09 | 27.57 | 36.67 |
| Proxy-W2V2-CTC-GCLB | 29.25 | 27.91 | 36.92 |
| Proxy-W2V2-CTC-D4AM | 28.46 | 26.89 | 35.33 |
| INIT | 30.98 | 29.61 | 38.61 |

signals. Furthermore, the WER value is calculated on a sample basis. In contrast, the conventional WER value is calculated using ground truth transcripts as targets and is derived by summing the errors across all testing samples.

By training our SE model with their pre-mixed training data, we executed our algorithm on the set with the following results:

Table 7: WER results on the L3DAS22 micro-A sets.

| | L3DAS22-dev-microA | | | L3DAS22-test-microA | | |
|---|---|---|---|---|---|---|
| method | TRAN | RNN | W2V2 | TRAN | RNN | W2V2 |
| NOISY | 31.14 | 60.80 | 38.85 | 27.59 | 57.98 | 35.70 |
| INIT | 22.93 | 35.65 | 28.00 | 21.94 | 33.72 | 26.67 |
| CLSO | 19.31 | 32.88 | 24.34 | 18.35 | 30.99 | 22.78 |
| SRPR | 19.03 | 31.82 | 23.68 | 18.08 | 29.95 | 22.54 |
| GCLB | 19.40 | 32.07 | 23.75 | 18.50 | 30.17 | 22.55 |
| D4AM | **18.68** | **31.61** | **23.59** | **17.88** | **29.71** | **22.19** |

## A.5 HUMAN PERCEPTION EVALUATION

Finally, we evaluated the enhanced speech signal using an objective perceptual metric, DNSMOS (Reddy et al., 2021). The DNSMOS value represents the perception score of human hearing, with higher scores indicating better quality. Table 8 shows the DNSMOS scores of enhanced speech produced by various SE methods. The real sets of CHiME-4 described in Section 4.2 were used as the testing data. Table 8 reveals that *D4AM* produced the highest DNSMOS score among all models trained with the ASR objective, and its DNSMOS score is marginally lower than that of *INIT* (only considering the regression objective for training). The result again substantiates our conjecture that any critical point of $\mathcal{L}_{cls}(\theta)$ should be "covered" by the critical point space of $\mathcal{L}_{reg}(\theta)$.

Table 8: DNSMOS results on the CHiME-4 real sets.

| method | NOIS | INIT | CLSO | SRPR | GCLB | D4AM |
|---|---|---|---|---|---|---|
| DNSMOS | 2.5574 | 3.2401 | 2.7665 | 3.0598 | 3.0398 | **3.1450** |

### A.6 VARIATION IN $\alpha_{srpr}$ DURING TRAINING

Fig. 3 shows the variation in $\alpha_{srpr}$ with and without gradient calibration applied over training steps. It is evident that the $\alpha_{srpr}$ values of both **SRPR** and **D4AM** converge rapidly, indicating that their learning process is stable.

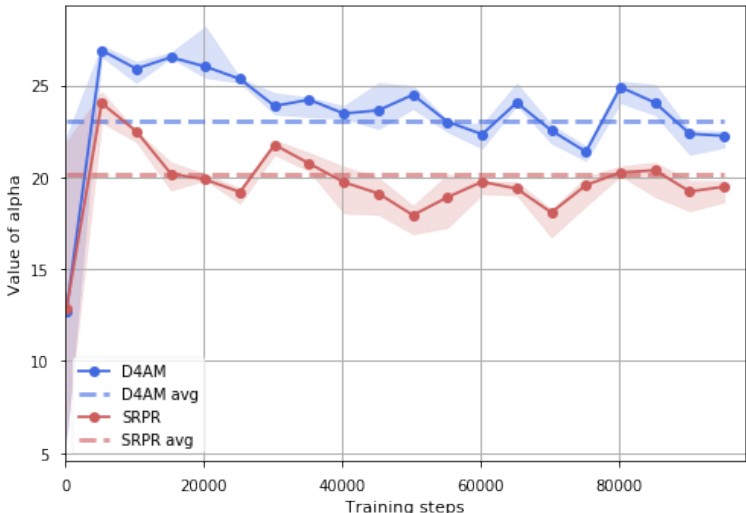

Figure 3: The change in the $\alpha_{srpr}$ value during training. The dashed lines represent the average $\alpha_{srpr}$ values.

### A.7 DERIVATION OF $\alpha_{gclb}$

To derive $\alpha_{gclb}$, the Lagrangian of the constrained optimization problem defined in Eq. 2 can be written as:

$$\mathcal{L}(g, \alpha) = \frac{1}{2}\|g - \nabla\mathcal{L}_{cls}^{\phi}(\theta^t)\|^2 - \alpha\langle g, \nabla\mathcal{L}_{reg}(\theta^t)\rangle \tag{6}$$

To determine the optimal value $g^* = \arg\min_g \mathcal{L}(g, \alpha)$, we set the derivatives of $\mathcal{L}(g, \alpha)$ with respect to $g$ to 0. By solving for $\nabla_g\mathcal{L}(\alpha, g) = 0$, we obtain $g = \nabla\mathcal{L}_{cls}^{\phi}(\theta^t) + \alpha \cdot \nabla\mathcal{L}_{reg}(\theta^t)$. Then, we can substitute $g = \nabla\mathcal{L}_{cls}^{\phi}(\theta^t) + \alpha \cdot \nabla\mathcal{L}_{reg}(\theta^t)$ into $\mathcal{L}(\alpha, g)$ and get the minimum $\mathcal{L}^*(\alpha)$:

$$\mathcal{L}^*(\alpha) = -\alpha\langle\nabla\mathcal{L}_{cls}^{\phi}(\theta^t), \nabla\mathcal{L}_{reg}(\theta^t)\rangle - \frac{1}{2}\alpha^2\|\nabla\mathcal{L}_{reg}(\theta^t)\|_2^2 + const \tag{7}$$

where $const$ is the summation of those terms regardless of $\alpha$. Here, the solution of $\alpha_{gclb}$ is the critical point of $\mathcal{L}^*(\alpha)$. Therefore, by deriving $\alpha$ such that $\nabla_\alpha\mathcal{L}^*(\alpha) = 0$, we can obtain the final result of $\alpha_{gclb}$:

$$\alpha_{gclb} = -[\![\langle\nabla\mathcal{L}_{cls}^{\phi}(\theta^t), \nabla\mathcal{L}_{reg}(\theta^t)\rangle < 0]\!] \cdot \frac{\langle\nabla\mathcal{L}_{cls}^{\phi}(\theta^t), \nabla\mathcal{L}_{reg}(\theta^t)\rangle}{\|\nabla\mathcal{L}_{reg}(\theta^t)\|_2^2} \tag{8}$$

