# OpenReview forum: "D4AM: A General Denoising Framework for Downstream Acoustic Models"
_ICLR.cc/2023/Conference — ICLR 2023 poster_

### Official Review · Reviewer_jXQS · 2022-10-23

**Confidence:** 4
**Correctness:** 3
**Technical Novelty And Significance:** 2
**Empirical Novelty And Significance:** 2
**Recommendation:** 5

**Clarity, Quality, Novelty And Reproducibility:**

I really enjoyed reading the paper. The writing was quite clear, and the language was quite precise. The attached material support reproducibility.

**Strength And Weaknesses:**

Strengths
1.Good literature review.
2. Insightful connection to related efforts in Section 3.
3. Evaluation across a variety of recognizers and datasets.
4. Helpful wav and spectrogram files in the supplementary section.
5. The attached source code promotes reproducibility.
6. The writing is very clear.

Growth opportunities
1. The authors claim that "tuning parameters in an ASR system is not a feasible solution for practical scenarios." With large-scale multi-condition training, it is not clear how much tuning is necessary.
2. The annotation cost comment on MCT is misleading. Typically, only the clean speech is annotated, incurring no additional cost for MCT.
3. What is the justification for the learning rate and update period values used? How sensitive is the solution to these parameters?
4. Why is the SNR range being investigated in the range from -4dB to 6dB?
5. CHIME-4 and Aurora-4 are medium vocabulary ASR tasks. It would be beneficial to evaluate the proposed approach on LVCSR tasks.
6. Only small improvement over GCLB is observed. On the Aurora-4 and Google ASR API on CHIME-4 tasks, even SRPR performs competitively.

**Summary Of The Paper:**

The paper proposes a denoising approach for speech enhancement for use in downstream tasks such as automatic speech recognition (ASR). They augment the typical end-to-end training with an auxiliary loss function that also seeks to minimize distortion with the notion of ASR-independent generalization. The results have been evaluated on 2 noisy ASR evaluation datasets and show a small improvement over a couple of previous systems.

**Summary Of The Review:**

The paper builds on a host of related work and shows small improvements on medium vocabulary ASR tasks and is computationally more efficient than a grid-search-based approach. Without LVCSR or large-scale MCT comparisons though, the power of the approach might need to come from its applicability to other tasks, e.g., intelligibility improvements of enhanced speech.

---

> ### Author Response · Authors · 2022-11-11
> **About LVCSR or large-scale MCT comparisons**
>
> We are grateful for Reviewer jXQS's advice. Before submitting our comments, we would like to humbly ensure what LVCSR and large-scale MCT models refer to since we believe our original experiment setups are based on LVCSR and large-scale MCT.
>
> For LVCSR, we have evaluated our performance on Aurora-4 in Table 1. Aurora-4 is recognized as an academically standardized and publicly available LVCSR. The official website of Aurora-4 (http://aurora.hsnr.de/aurora-4.html) also states that Aurora-4 is a "large vocabulary task."
> For large-scale MCT ASR models, we used Google API, considered a large-scale MCT ASR model. The results of testing the proposed D4AM on Google API have been listed in Table 3. We believe that in the experiments of Table 3, the testing data is a "large vocabulary task," and the ASR is Google API. The overall setup should belong to an LVCSR and large-scale MCT setup.
>
> In addition, we also posted the results tested on LibriSpeech's test-clean set and their noisy versions in Appendix Table 5. LibriSpeech is also widely considered as an LVCSR test set. The website (https://www.openslr.org/12) states that LibriSpeech is a large-scale (1000-hours) corpus. Thus, all the used datasets for training and testing were regarded as large-scale datasets in academics, and the evaluated MCT models could also be considered large-scale.
>
> It would be greatly appreciated if you could provide us with some dataset references so that we can conduct further research on a large scale.

---

> ### Author Response · Authors · 2022-11-17
> **To Reviewer jXQS: Page 1**
>
> We really appreciate Reviewer jXQS's advice. We have revised the corresponding content of Section 2 for more clear descriptions after carefully reading your comments. Issues that need to be clarified are listed below.
>
> ---
> > 1. The authors claim that "tuning parameters in an ASR system is not a feasible solution for practical scenarios." With large-scale multi-condition training, it is not clear how much tuning is necessary.
>
> i. It has been shown that fine-tuning ASR models can achieve better performance even under multi-condition training (Prasad et al., 2021 [1]). However, tuning parameters in an ASR system is not a viable solution for real scenarios due to the complexity or inaccessibility of ASR models. Moreover, **we would like to emphasize that our goal is not to discuss “how much tuning is necessary” a certain MCT model needs but rather to ensure that unseen ASR models can adequately handle inputs in all situations (regardless of ASR’s noise robustness)**. \
> ii. Google API can be viewed as a large-scale MCT model. In Table 3, we have clearly shown that Google API suffers performance degradation when fed directly with noisy speech input, various denoising methods provide performance improvements, and our method performs best.
>
> [1] Archiki Prasad, Preethi Jyothi, Rajbabu Velmurugan, “An Investigation of End-to-End Models for Robust Speech Recognition” in Proc. ICASSP 2021
>
> ---
> > 2. The annotation cost comment on MCT is misleading. Typically, only the clean speech is annotated, incurring no additional cost for MCT.
>
> Thank you for the comment. We do know that there is no additional cost in preparing the MCT set as long as the annotations are available for a clean training set. **Please note that the original text is "Furthermore, ASR is usually trained with a large amount of speech-text pairs, requiring expensive annotation costs.". We do not mention "MCT" here**, and the annotation cost is not related to MCT training. Annotated speech can be used to produce both noisy-clean pairs and (noisy) speech-text pairs; unannotated speech can be used to produce noisy-clean pairs. Practically, there is much more unannotated speech than annotated speech, so the availability of noisy-clean pairs is much better.
>
> ---
> > 3. What is the justification for the learning rate and update period values used? How sensitive is the solution to these parameters?
>
> The learning rate and update period are hyper settings in our framework. According to our experiments, if the clipping operation is applied, they do not affect the trend of the results but only slightly affect the convergence speed of the coefficients.
>
> ---
> > 4. Why is the SNR range being investigated in the range from -4dB to 6dB?
>
> For speech enhancement and robust ASR, low SNR cases are the focus of processing. Typically, the SNR range needs to span both positive and negative values. Based on our previous experience, the lower and upper bounds can be modified. As long as the SNR range spans 0, the trend of the results will be similar.
>
> ---
> > 5. Only small improvement over GCLB is observed. On the Aurora-4 and Google ASR API on CHIME-4 tasks, even SRPR performs competitively.
>
> i. Our joint training scheme aims to "regularize" the optimization process so that the enhanced results can be applied to other unseen ASR models. **The role of the regularizer is to ensure consistent improvements across situations rather than focusing on how much improvement can be achieved for a certain ASR**. \
> ii. Our experiments cover eight evaluation subsets and five evaluated ASR systems (40 situations in total). Our proposed method can generally bring improvements in these situations. We believe that the effectiveness of our proposed method has been well verified.

---

> ### Author Response · Authors · 2022-11-17
> **To Reviewer jXQS: Page 2**
>
> > 6. Without LVCSR or large-scale MCT comparisons though, the power of the approach might need to come from its applicability to other tasks, e.g., intelligibility improvements of enhanced speech.
>
> i. We would like to humbly ask what LVCSR and large-scale MCT models refer to, since we believe our experiment setups are based on LVCSR and large-scale MCT. \
> For LVCSR, we have evaluated our performance on Aurora-4 in Table 1. Aurora-4 is recognized as an academically standardized and publicly available LVCSR task. The official website of Aurora-4 (http://aurora.hsnr.de/aurora-4.html) also states that Aurora-4 is a "large vocabulary task." \
> For large-scale MCT, we have used Google API, considered a large-scale MCT ASR model. Table 3 shows the test results on the Google API. The test data comes from a "large vocabulary task," and the ASR model of Google API is definitely a large-scale MCT model. \
> In addition, we have also reported the results tested on LibriSpeech's test-clean set and its noisy versions in Table 5 in Appendix. LibriSpeech is also widely regarded as a LVCSR task. The website (https://www.openslr.org/12) states that LibriSpeech is a large-scale (1000-hours) corpus. Therefore, all datasets used for training and testing in this work are considered large-scale datasets in academia, and the evaluated MCT models can also be considered large-scale. It would be greatly appreciated if you could provide information on other publicly available datasets so that we can conduct further large-scale research.
>
> ii. We would like to clarify that our method aims to improve ASR performance. Any improvement in the STOI score is indicative only, as we cannot really determine the relationship between WER and STOI. We have demonstrated the DNSMOS results on the CHIME-4 real set. D4AM scored second only to INIT among all speech enhancement models. The STOI results on the Aurora-4 evaluation sets are as follows:
> | |  dev-wv1   | dev-wv2  | test-wv1  | test-wv2  |
> |  ----  |  ----  | ----  | ----  | ----  |
> | NOIS  | 0.8931 | 0.7918 | 0.9022 | 0.7708 |
> | INIT | 0.9618 | 0.8718 | 0.9661 | 0.8293 |
> | CLSO | 0.9327 | 0.8329 | 0.9385 | 0.7940 |
> | SRPR | 0.9542 | 0.8644 | 0.9590 | 0.8237 |
> | GCLB | 0.9535 | 0.8626 | 0.9584 | 0.8219 |
> | D4AM | 0.9584 | 0.8702 | 0.9629 | 0.8299 |

---

### Official Review · Reviewer_mD2F · 2022-10-24

**Confidence:** 2
**Correctness:** 4
**Technical Novelty And Significance:** 3
**Empirical Novelty And Significance:** 3
**Recommendation:** 6

**Clarity, Quality, Novelty And Reproducibility:**

The paper is generally clearly written and easy to follow, but some parts could be improved:
 - In Section 2, the second half is not easy to read as it's not really clear what is related work, what is motivation and what is the proposed approach. Using Subsections should help, as well as are clearly stating which part of the section (typically Equation (1) and (2)) are related work and which part are original contributions.
- In Section 4, using bold font for the acronyms is quite distracting, using italic (or \emph) should make is more readable.
- Section 4.1, 3rd paragraph: "SRPR, GCLB, and D4AM are associated with Chen et al. (2015); Ma et al. (2021).": I thought the D4AM is the proposed approach, why is it "associated" with other papers? is it a typo?

The proposed approach seems novel, although it's unclear which part is original (see Strengths and Weaknesses).

**Strength And Weaknesses:**

Strengths:
 - The approach presented in the paper is significant as it is designed for unseen ASR systems and yields state-ot-the-art performance.
 - The adjustment scheme is very interesting and makes the approach very practical and appealing to use.
 - The experiments are thorough and clearly show the viability of the proposed approach.

Weaknesses:
 - The paper doesn't situate well the proposed approach with the literature: in the third paragraph of Section 2, several related works are cited which seems to be very similar to the proposed approach as they all use the auxiliary task approach, but it's unclear which tasks and how it is relating to the proposed approach. Please describe the relevant related works and clarify.
- The clarity of the paper could be improved.

**Summary Of The Paper:**

This paper presents a method for Speech Enhancement (SE) applied to Automatic Speech Recognition (ASR). In the proposed approach, the SE model is trained to enhance noisy speech and to keep improving recognition performance of the acoustic model. This is accomplished by training the SE model with multi-task learning, where one of the losses is an estimation of the ASR performance, together with an adjustment scheme to learn the weights directly. The approach is evaluated on several noisy ASR benchmarks, including an unseen ASR system and is shown to improve peformance.

**Summary Of The Review:**

Overall, the approach seems novel and the findings are significant. The domain is very specific (using SE for ASR) and might not be of huge interest to the whole ICLR community, but the algorithm with the adjustment could be useful and is worth publishing. I thus recommend to accepting the paper.

---

> ### Author Response · Authors · 2022-11-17
> **To Reviewer mD2F: Page 1**
>
> We really appreciate Reviewer mD2F's advice. We have revised the corresponding content of Sections 1 and 2 according to the suggestions and modified the font in Section 4. The issues that need to be clarified are listed below.
>
> ---
> > 1. The paper doesn't situate well the proposed approach with the literature: in the third paragraph of Section 2, (1) several related works are cited which seems to be very similar to the proposed approach as they all use the auxiliary task approach, but it's unclear which tasks and how it is relating to the proposed approach. Please describe the relevant related works and clarify. The clarity of the paper could be improved.
>
> In Section 2, our original intention is to emphasize that a regression objective can be used as an auxiliary objective. ARML [1] is a framework that can theoretically satisfy our scenario: the critical points of the main task are also covered by the critical point space of the auxiliary loss, and other auxiliary task approaches are related works. In auxiliary learning, the main task and auxiliary tasks usually share the same module (one stage). In our scenario, we need to derive the main task objective by cascading an additional proxy module, and our goal is to extend the conventional auxiliary framework into a “two-stage” audio processing application (the first stage is a speech denoiser, and the second stage is an ASR model).
>
> (1) For audio processing applications, we develop a coefficient search scheme that achieves good performance with only one training session (no need to train multiple models to find the best one). \
> (2) For auxiliary learning, auxiliary learning frameworks typically focus on a one-stage module scenario (all tasks share the same module). In contrast, **we develop a modified regularizer that extends the scenario to “two-stage” and adapts the results of the first stage to other non-trained second-stage modules**.
>
> Our technical novelties are mainly presented in Sections 3.2 and 3.3. In Section 3.2, we choose the regression gradient as a constraint to calibrate the classification objective. This is a new strategy to solve the problem of not being able to access the real second-stage objective (only a proxy one is feasible). In Section 3.3, we develop an improved version of the ARML implementation to better stabilize the coefficient adjustment process.
>
> For clarity, in the revised paper, we have changed the title of Section 2 to Motivation and Main Idea, revised some sentences and divided Section 2 into two subsections with headings.
>
> ---
> > 2. In Section 2, the second half is not easy to read as it's not really clear what is related work, is motivation and what is the proposed approach. Using Subsections should help, as well as are clearly stating which part of the section (typically Equation (1) and (2)) are related work and which part are original contributions.
>
> Thank you for the suggestions. After reading Section 2 carefully, we see that perhaps the biggest source of misunderstanding is its title. Section 2 mainly expounds on our research motivation and main idea. Along the way, we need to explain some past related work and which existing methods we have applied. To avoid confusion, in the revised paper, we have changed the title of Section 2 to Motivation and Main Idea, revised some sentences, and divided it into two subsections with headings.
>
> ---
> > 3. Section 4.1, 3rd paragraph: "SRPR, GCLB, and D4AM are associated with Chen et al. (2015); Ma et al. (2021).": I thought the D4AM is the proposed approach, why is it "associated" with other papers? is it a typo?
>
> We apologize for this unclear description. Our intention is to describe that these previous works also combine ASR and speech enhancement (regression) losses for model training. They have investigated the effectiveness of joint training with predefined coefficients, whereas our proposed approach automatically determines the weights by means of parameter search. In the revised version uploaded, we have revised these sentences.

---

### Official Review · Reviewer_DCcQ · 2022-10-29

**Confidence:** 4
**Correctness:** 3
**Technical Novelty And Significance:** 2
**Empirical Novelty And Significance:** 2
**Recommendation:** 5

**Clarity, Quality, Novelty And Reproducibility:**

Clarity
- It is difficult to read Section 3, especially Section 3.1, due to many inline equations. It is better to provide more high-level derivations; some details can be moved to the appendix sections.

Quality
- I cannot fully agree with the "SE approaches do not consider the generalization ability to unseen ASR systems." I think it is valid for weak speech enhancement models, but the latest state-of-the-art speech enhancement shows the generalized performance without changing the ASR system, e.g., the L3DAS challenge https://www.l3das.com/icassp2022/index.html fixes the ASR backend, but the many speech enhancement systems improve the ASR performance without using the ASR backend as a loss function. I understand the claim of "unseen ASR systems" to some extent, but some strong speech enhancement may achieve such properties.
- The effectiveness of the automatic balancing scheme compared with the grid search is marginal. To show the effectiveness of the proposed method, it may find more complex cases for the grid search (e.g., more tuning parameters).

Novelty
- The optimization of speech enhancement with ASR loss is not novel. Of course, a part of dealing with unseen ASR systems is novel, but it is not sufficiently novel, and the paper requires more clarification of the technical novelty of this paper.
- I think the automatic balancing scheme seems to be novel.

Reproducibility
- Training data seems to be based on their own simulation methods. It is better to release this setup for reproducibility

**Strength And Weaknesses:**

Strength
- consistent improvements of the ASR performance even with a black box ASR API. The problem setup of developing universal speech enhancement techniques for various ASR systems is crucial.
- automatic balancing scheme of regression and classification losses seem to be novel
- showing the robustness of the balancing scheme experimentally.

Weaknesses
- optimization of speech enhancement with ASR loss is not novel. Of course, a part of dealing with unseen ASR systems is novel, but it is not a sufficient novelty.
- the method is specific to speech processing problems and would not attract general machine learning and AI researchers. The automatic balancing scheme seems to be a general scheme, and if this has more critical or different applications, this part will be improved more.
- Although the balancing scheme is sophisticated, the result is not very different from a conventional grid search. It is not difficult to tune a single hyper-parameter with the dev set.

**Summary Of The Paper:**

This paper proposes a denoising frontend for automatic speech recognition. The method combines speech enhancement (regression) and recognition (classification) loss. The paper presents a scheme for automatically balancing the regression and classification losses. The paper shows the effectiveness of the proposed method in the efficacy of noisy speech recognition experiments.

**Summary Of The Review:**

The paper does not have sufficient novelty in terms of the technical/algorithm part. Various researchers have already studied the joint training of both enhancement and recognition losses (as shown in Section 2). The automatic balancing scheme seems novel, but the effectiveness compared with the grid search is marginal.

Other suggestions
- Please discuss the reverberation cases. This is very critical for actual ASR deployment.
- Can we use this method without the regression loss? In the practical scenario (real data situations), we cannot prepare the clean data, and we cannot construct the regression loss. I want the authors to discuss such cases and how the proposed method works.
- Abstract: It is better to provide more concrete descriptions (e.g., with some numbers) to claim the effectiveness of the proposed method experimentally.
- Can you discuss whether the proposed automatic balancing scheme more generally applies to the other machine learning problems? I think such a discussion makes the potential effectiveness of this scheme strong.
- The introduction (section 1) and motivation (section 2) have some similar logical flows, and this part can be refined.
- Why did you use DEMUCS? Can you add some references about the state-of-the-art performance of DEMUCS in some SE tasks?
- Table 1. Please use the same decimal place for all numbers (2nd decimal place).
- Section 4.2 "As seen in Table 2, the result of INIT is comparable to the state-of-the-art results of CHiME-4" Can you add a reference? Which system are you comparing?

---

> ### Author Response · Authors · 2022-11-17
> **To Reviewer DCcQ: Page 1**
>
> We really appreciate Reviewer DCcQ's advice. We have revised the corresponding content of Abstract and Sections 1, 3, and 4 according to the suggestions. Issues that need to be clarified are listed below.
>
> ---
> > 1. the method is specific to speech processing problems and would not attract general machine learning and AI researchers. The automatic balancing scheme seems to be a general scheme, and if this has more critical or different applications, this part will be improved more.
>
> i. We would like to emphasize that audio is an important sequential data task to study, and in ICLR, the application of machine learning to audio is included in the list of relevant topics. Moreover, speech processing itself is an important topic in the AI community. Several audio application papers are published in ICLR every year [1-18]. \
> ii. Due to limited pages, we only evaluated our algorithm on the SE-ASR task, but our framework may be applicable to other non-sequential data tasks. For example, in computer vision, super-resolution and image inpainting are two-stage module scenarios that can be trained using synthetic source-target pairs. As long as the critical points of the main task are covered by the critical point space of the auxiliary loss, the same auxiliary learning scheme can be applied.
>
> [1] Hyeong-Seok Choi, Jang-Hyun Kim, Jaesung Huh, Adrian Kim, Jung-Woo Ha, Kyogu Lee, "PHASE-AWARE SPEECH ENHANCEMENT WITH DEEP COMPLEX U-NET" in Proc. ICLR, 2019 \
> [2] Chris Donahue, Julian McAuley, Miller Puckette, “Adversarial Audio Synthesis” in Proc. ICLR, 2019 \
> [3] Jesse Engel, Kumar Krishna Agrawal, Shuo Chen, Ishaan Gulrajani, Chris Donahue, Adam Roberts, “GANSynth: Adversarial Neural Audio Synthesis” in Proc. ICLR, 2019 \
> [4] Raza Habib, Soroosh Mariooryad, Matt Shannon, Eric Battenberg, R. J. Skerry-Ryan, Daisy Stanton, David Kao, Tom Bagby, “Semi-Supervised Generative Modeling for Controllable Speech Synthesis” in Proc. ICLR, 2020 \
> [5] Alexei Baevski, Steffen Schneider, Michael Auli, “vq-wav2vec: Self-Supervised Learning of Discrete Speech Representations” in Proc. ICLR, 2020 \
> [6] Zhoutong Zhang, Yunyun Wang, Chuang Gan, Jiajun Wu, Joshua B. Tenenbaum, Antonio Torralba, William T. Freeman, “Deep Audio Priors Emerge From Harmonic Convolutional Networks” in Proc. ICLR, 2020 \
> [7] Alexander Richard, Dejan Markovic, Israel D. Gebru, Steven Krenn, Gladstone Alexander Butler, Fernando De la Torre, Yaser Sheikh, "Neural Synthesis of Binaural Speech From Mono Audio" in Proc. ICLR, 2021 \
> [8] Yi Ren, Chenxu Hu, Xu Tan, Tao Qin, Sheng Zhao, Zhou Zhao, Tie-Yan Liu, “FastSpeech 2: Fast and High-Quality End-to-End Text to Speech” in Proc. ICLR 2021 \
> [9] Jiahui Yu, Wei Han, Anmol Gulati, Chung-Cheng Chiu, Bo Li, Tara N Sainath, Yonghui Wu, Ruoming Pang, “Dual-mode ASR: Unify and Improve Streaming ASR with Full-context Modeling” in Proc. ICLR, 2021 \
> [10] Jeff Donahue, Sander Dieleman, Mikolaj Binkowski, Erich Elsen, Karen Simonyan, “End-to-end Adversarial Text-to-Speech" in Proc. ICLR, 2021 \
> [11] Mingjian Chen, Xu Tan 0003, Bohan Li, Yanqing Liu, Tao Qin, Sheng Zhao, Tie-Yan Liu, “AdaSpeech: Adaptive Text to Speech for Custom Voice” in Proc. ICLR, 2021 \
> [12] Kyuhong Shim, Jungwook Choi, Wonyong Sung, “Understanding the Role of Self Attention for Efficient Speech Recognition” in Proc. ICLR, 2022 \
> [13] Mia Chiquier, Chengzhi Mao, Carl Vondrick, “Real-Time Neural Voice Camouflage” in Proc. ICLR, 2022 \
> [14] Wenyong Huang, Zhenhe Zhang, Yu Ting Yeung, Xin Jiang, Qun Liu, “SPIRAL: Self-supervised Perturbation-Invariant Representation Learning for Speech Pre-Training” in Proc. ICLR, 2022 \
> [15] Shaojin Ding, Tianlong Chen, Zhangyang Wang, “Audio Lottery: Speech Recognition Made Ultra-Lightweight, Noise-Robust, and Transferable” in Proc. ICLR, 2022 \
> [16] Max W. Y. Lam, Jun Wang, Dan Su, Dong Yu, “BDDM: Bilateral Denoising Diffusion Models for Fast and High-Quality Speech Synthesis” in Proc. ICLR, 2022 \
> [17] Hadi Abdullah, Aditya Karlekar, Vincent Bindschaedler, Patrick Traynor, “Demystifying Limited Adversarial Transferability in Automatic Speech Recognition Systems” in Proc. ICLR, 2022 \
> [18] Max Morrison, Rithesh Kumar, Kundan Kumar, Prem Seetharaman, Aaron Courville, Yoshua Bengio, “Chunked Autoregressive GAN for Conditional Waveform Synthesis” in Proc. ICLR, 2022

---

> ### Author Response · Authors · 2022-11-17
> **To Reviewer DCcQ: Page 2**
>
> > 2. Although the balancing scheme is sophisticated, the result is not very different from a conventional grid search. It is not difficult to tune a single hyper-parameter with the dev set.
>
> The focus of this study is to find appropriate coefficients “efficiently.” **In other words, the goal is to train our SE model to achieve good performance “without paying extra training costs.”** In our experiments, training a model with one specific coefficient **requires about 20 GB of memory storage and 5 days of training time**. The optimal coefficients are different for different loss types and datasets, making it **difficult to perform a time-consuming grid search every time**. Although only one hyper-parameter needs to be tuned in the current task, our framework can be extended to multi-coefficient scenarios with appropriate modifications. Experimental results show that the model trained by our method is comparable to the best model selected by grid search. Since our method has large computational advantages, it is very useful for research (academic) teams with limited computational resources.
>
> ---
> > 3. I cannot fully agree with the "SE approaches do not consider the generalization ability to unseen ASR systems." I think it is valid for weak speech enhancement models, but the latest state-of-the-art speech enhancement shows the generalized performance without changing the ASR system, e.g., the L3DAS challenge https://www.l3das.com/icassp2022/index.html fixes the ASR backend, but the many speech enhancement systems improve the ASR performance without using the ASR backend as a loss function. I understand the claim of "unseen ASR systems" to some extent, but some strong speech enhancement may achieve such properties.
>
> i. We are very grateful to reviewer DCcQ for pointing out this overclaiming issue. We agree that our original sentences may have been too arbitrary and have revised them in the revised paper. We rephrased the sentence “However, the training objectives of existing SE approaches do not consider the generalization ability to unseen ASR systems.” in Abstract as “However, the training objectives of existing SE approaches do not effectively integrate speech-text and noisy-clean paired data for training.” \
> ii. We agree that a well-searched integration of architectures and objectives can lead to satisfactory performance. Since our method can be combined with existing SOTA systems, in our experiments, we only change the training objective and keep other settings unchanged.
>
> ---
> > 4. The effectiveness of the automatic balancing scheme compared with the grid search is marginal. To show the effectiveness of the proposed method, it may find more complex cases for the grid search (e.g., more tuning parameters).
>
> Our goal is to train our SE model to achieve good performance “without paying additional training costs.” Strictly speaking, the grid search results are **NOT our baseline results but top-line results**. Our experimental results show that the model trained by our method is comparable to the best model selected by the time-consuming grid search. Due to its large computational advantage, our method is very useful for research (academic) teams with limited computational resources. Although only one hyper-parameter needs to be tuned in the current task, our framework can be extended to multi-coefficient scenarios with appropriate modifications. We will leave this as future work.

---

> ### Author Response · Authors · 2022-11-17
> **To Reviewer DCcQ: Page 3**
>
> > 5. The optimization of speech enhancement with ASR loss is not novel. Of course, a part of dealing with unseen ASR systems is novel, but it is not sufficiently novel, and the paper requires more clarification of the technical novelty of this paper.
>
> i. Indeed, optimizing speech enhancement using an ASR loss is not novel. Our novelty lies in the development of a coefficient search scheme combining ASR loss and speech enhancement loss. Several successful auxiliary learning methods have been proposed in the computer vision area, most of which are listed in the motivation part of Section 2. We believe that developing such a coefficient search scheme is novel enough, especially in audio processing applications. \
> (1) For audio processing applications, we develop a coefficient search scheme that achieves good performance with only one training session (no need to train multiple models to find the best one). \
> (2) For auxiliary learning, auxiliary learning frameworks typically focus on a one-stage module scenario (all tasks share the same module). In contrast, **we develop a modified regularizer that extends the scenario to “two-stage” and adapts the results of the first stage to other non-trained second-stage modules**.
>
> ii. Our technical novelties are mainly presented in Sections 3.2 and 3.3. In Section 3.2, we choose the regression gradient as a constraint to calibrate the classification objective. This is a new strategy to solve the problem of not being able to access the real second-stage objective (only a proxy one is feasible). In Section 3.3, we develop an improved version of the ARML implementation to better stabilize the coefficient adjustment process.
>
> ---
> > 6. Training data seems to be based on their own simulation methods. It is better to release this setup for reproducibility
>
> **Please note that the simulation method has been included in the supplementary material. All details are documented in data.py.** The mixing procedure is described in Section 4.1. The mixing process is conducted during data loading. For noisy-clean pairs, each segment is truncated or duplicated to increase sample complexity.
>
> ---
> > 7. Please discuss the reverberation cases. This is very critical for actual ASR deployment.
>
> We agree that the discussion of convolutional interference is critical for actual ASR deployment. Most of our experimental settings are cross-dataset or cross-device scenarios, which means that at least one kind of convolutional interference has been considered. We have further executed our algorithm on the L3DAS22 micro-A set with the following results:
>
> L3DAS22_Task1_dev-microA:
> | |  TRAN | RNN  | W2V2 |
> |  ----  |  ----  | ----  | ----  |
> | NOIS  | 31.14 | 60.80 | 38.85 |
> | INIT | 22.93 | 35.65 | 28.00 |
> | CLSO | 19.31 | 32.88 | 24.34 |
> | SRPR | 19.03 | 31.82 | 23.68 |
> | GCLB | 19.40 | 32.07 | 23.75 |
> | D4AM | **18.68** | **31.61** | **23.59** |
>
> L3DAS22_Task1_test-microA:
> | |  TRAN | RNN  | W2V2 |
> |  ----  |  ----  | ----  | ----  |
> | NOIS  | 27.59 | 57.98 | 35.70 |
> | INIT | 21.94 | 33.72 | 26.67 |
> | CLSO | 18.35 | 30.99 | 22.78 |
> | SRPR | 18.08 | 29.95 | 22.54 |
> | GCLB | 18.50 | 30.17 | 22.55 |
> | D4AM | **17.88** | **29.71** | **22.19** |
>
> The results also validate the effectiveness of D4AM under reverberation. The results have been updated in the Appendix section, and the corresponding checkpoints have been uploaded to the drive link (https://mega.nz/file/deEzzZ5Q#Y53kTcfuAeke5NmE96yiFac_RnQTg-Be8Z_FmxgUag8) for reproducibility.
> Please note that the scenario of L3DAS22 is different from ours: \
> i. The WER definition of L3DAS22 is different from the regular version. **In L3DAS22, the reference target for computing WER is NOT a true transcription**. The WER is the word-level distance between the ASR model’s predicted transcripts for the clean and enhanced signals. In addition, the WER value is calculated on a sample basis. In contrast, the conventional WER value (ours) is calculated using ground truth transcripts as targets and is derived by summing the errors across all testing samples. \
> ii. The SOTA systems of L3DAS22 usually use multi-stage processing. In addition to working on training schemes, they may employ some specialized pre- and post-processing techniques, which are also critical to the performance. \
> iii. The input of L3DAS22 is a multi-channel signal. Participants can utilize the spatial relationship between channels to achieve better performance. In our experiments, **we focus on the single-channel SE task and treat different channels as separate signals for training and testing**.

---

> ### Author Response · Authors · 2022-11-17
> **To Reviewer DCcQ: Page 4**
>
> > 8. Can we use this method without the regression loss? In the practical scenario (real data situations), we cannot prepare the clean data, and we cannot construct the regression loss. I want the authors to discuss such cases and how the proposed method works.
>
> A feasible solution to handle real data situations is to integrate real data with simulated data. During training, both real and simulated speech-text pairs can be used to generate the classification loss, and the simulated noisy-clean pairs can be used to generate the auxiliary (regression) loss.
>
> ---
>
> > 9. Can you discuss whether the proposed automatic balancing scheme more generally applies to the other machine learning problems? I think such a discussion makes the potential effectiveness of this scheme strong.
>
> Our proposed method can be applied to other speech-processing applications, such as speaker recognition, and is believed to be applicable to other sequential data tasks (such as ECG signals). In computer vision, super-resolution and image inpainting are two-stage module scenarios that can be trained using synthetic source-target pairs. As long as the critical points of the main task are covered by the critical point space of the auxiliary loss, the same auxiliary learning scheme can be applied.
>
> ---
> > 10. The introduction (section 1) and motivation (section 2) have some similar logical flows, and this part can be refined.
>
> Although Sections 1 and 2 overlap somewhat for the sake of fluency, we would like to emphasize that most of the descriptions in these two sections are different.
> In Section 1, we highlight the background of speech enhancement and point out the differences between our proposed method and existing methods.
> In Section 2, we first additionally illustrate the advantages of applying speech enhancement as an ASR pre-processor and then explain our main idea of applying auxiliary learning to make speech enhancement models provide better performance for unseen ASR models.
> To avoid confusion, in the revised paper, we have changed the title of Section 2 and added two subtitles to Section 2.
>
> ---
> > 11. Why did you use DEMUCS? Can you add some references about the state-of-the-art performance of DEMUCS in some SE tasks?
>
> In the work of Prasad et al. in 2021 [1], DEMUCS mostly outperforms other architectures in the robust ASR investigation. Furthermore, the implementation of DEMUCS is publicly available and can be easily extended.
> For SOTA information on speech enhancement, please refer to the official website of DNS-Challenge 2020 (https://www.microsoft.com/en-us/research/academic-program/deep-noise-suppression-challenge-interspeech-2020/results/)
> Their institute is “Facebook AI, INRIA”.
>
> [1] Archiki Prasad, Preethi Jyothi, Rajbabu Velmurugan, “An Investigation of End-to-End Models for Robust Speech Recognition” in Proc. ICASSP 2021
>
> ---
> > 12. Section 4.2 "As seen in Table 2, the result of INIT is comparable to the state-of-the-art results of CHiME-4" Can you add a reference? Which system are you comparing?
>
> We are sorry for the unclear description. Here, our original intention is to describe that the results of INIT reach the same scale in other papers. However, note that this is not a fair comparison because our training data and other experimental settings (such as trainable ASR models or the training details of SE models) are different from other SOTA papers.
>
> For those SOTA references, please refer to: \
> [1] CHiME4’s official website (https://spandh.dcs.shef.ac.uk/chime_challenge/CHiME4/results.html) \
> [2] Yen-Ju Lu, Xuankai Chang, Chenda Li, Wangyou Zhang, Samuele Cornell, Zhaoheng Ni, Yoshiki Masuyama, Brian Yan, Robin Scheibler, Zhong-Qiu Wang, Yu Tsao, Yanmin Qian, Shinji Watanabe, “ESPnet-SE++: Speech Enhancement for Robust Speech Recognition, Translation, and Understanding” in Proc. Interspeech 2022 \
> [3] Tsubasa Ochiai, Shinji Watanabe, Takaaki Hori, John R. Hershey, “Multichannel End-to-end Speech Recognition” in Proc. ICML 2017

---

> ### Comment · Reviewer_DCcQ · 2022-11-29
> **Thanks for the detailed responses.**
>
> Thanks for the detailed responses and a lot of clarifications about my comments.
> I could not entirely change my mind regarding the novelty and effectiveness, but the paper's readability would be significantly improved based on the update. I particularly agree that the automatic balancing of regression and classification is beneficial for various speech-processing applications. I increased my score.

---

### Author Response · Authors · 2022-12-12
**Concluding Remarks**

The authors thank all Reviewers for their invaluable comments. Efforts of the Reviewers and the Chairs towards improving the manuscript are deeply appreciated. Also, we really appreciate all the reviewer’s revision suggestions, and we have accordingly followed all the to modify our descriptions. Because of the reaching of the discussion end, we would like to briefly highlight our work for your reference. To sum up all the comments, we would like to list three key points:

(1) (To Reviewer-DCcQ/Reviewer-mD2F/Reviewer-jXQS) Our goal is to extend the conventional auxiliary framework into a “two-stage” audio processing application (the first stage is a speech denoiser, and the second stage is an ASR model) and, thus provide better generalization abilities to other second-stage (ASR) modules that are non-trained.

(2) (To Reviewer-DCcQ/Reviewer-jXQS) Our joint training scheme aims to "regularize" the optimization process so that the enhanced results can be applied to other unseen ASR models, which is noveland the main contribution of this work. The role of the regularizer is to ensure consistent improvements across situations rather than focusing on how much improvement can be achieved for a certain ASR. Our experiments cover eight evaluation subsets and five evaluated ASR systems (40 situations in total). Our proposed method can generally bring improvements in these situations. We believe that the effectiveness of our proposed method has been well verified.

(3) (To Reviewer-jXQS) We believe that our original experiment setups are based on LVCSR and large-scale MCT. For LVCSR, the official websites of Aurora-4 and LibriSpeech both state that they are "large vocabulary tasks." For large-scale MCT, we have used Google API, considered a large-scale MCT ASR model.

---

### Decision · Program_Chairs · 2023-01-20

**Decision:**

Accept: poster

**Justification For Why Not Higher Score:**

* novelty should be much better

**Justification For Why Not Lower Score:**

* interesting results

**Metareview: Summary, Strengths And Weaknesses:**

This paper is about denoising for ASR.

Strengths:
* good results
* adjustment scheme is interesting
* clear writing

Weaknesses:
* novelty is lacking

**Note From Pc:**

if the above contains the word "oral" or "spotlight" please see: "oral" presentation means -> notable-top-5% and "spotlight" means -> notable-top-25%. As stated in our emails, we are disassociating presentation type from AC recommendations